# Suicide in Rural Australia: Are Farming-Related Suicides Different?

**DOI:** 10.3390/ijerph17062010

**Published:** 2020-03-18

**Authors:** Alison Kennedy, Jessie Adams, Jeremy Dwyer, Muhammad Aziz Rahman, Susan Brumby

**Affiliations:** 1National Centre for Farmer Health, School of Medicine, Deakin University, Victoria 3300, Australia; aziz.rahman@deakin.edu.au (M.A.R.); susan.brumby@wdhs.net (S.B.); 2Western District Health Service, Foster Street, Hamilton, Victoria 3300, Australia; jessie.adams@wdhs.net; 3Coroners Court of Victoria, Department of Forensic Medicine, Monash University, Victoria 3006, Australia; jeremy.dwyer@coronerscourt.vic.gov.au; 4School of Nursing and Midwifery, LaTrobe University, Victoria 3086, Australia; 5School of Nursing and Healthcare Professions, Federation University, Victoria 3806, Australia

**Keywords:** suicide, mental health, risk factors, farmers, rural population, Australia

## Abstract

Rural Australians experience a range of health inequities—including higher rates of suicide—when compared to the general population. This retrospective cohort study compares demographic characteristics and suicide death circumstances of farming- and non-farming-related suicides in rural Victoria with the aim of: (a) exploring the contributing factors to farming-related suicide in Australia’s largest agricultural producing state; and (b) examining whether farming-related suicides differ from suicide in rural communities. Farming-related suicide deaths were more likely to: (a) be employed at the time of death (52.6% vs. 37.7%, OR = 1.84, 95% CIs 1.28–2.64); and, (b) have died through use of a firearm (30.1% vs. 8.7%, OR = 4.51, 95% CIs 2.97–6.92). However, farming-related suicides were less likely to (a) have a diagnosed mental illness (36.1% vs. 46.1%, OR=0.66, 95% CIs 0.46–0.96) and, (b) have received mental health support more than six weeks prior to death (39.8% vs. 50.0%, OR = 0.66, 95% CIs 0.46–0.95). A range of suicide prevention strategies need adopting across all segments of the rural population irrespective of farming status. However, data from farming-related suicides highlight the need for targeted firearm-related suicide prevention measures and appropriate, tailored and accessible support services to support health, well-being and safety for members of farming communities.

## 1. Introduction

Suicide represents approximately 1.5% of all global deaths and is the 7th leading cause of death in Australasia. [1]. In Australia, suicide is the 14th leading cause of death, yet accounts for the highest number of years of potential life lost [2]. Suicide rates vary across different regions of Australia, with the rates increasing outside of capital cities [3].

Almost one-third of Australians live outside of capital cities [4] and face a range of health inequities—in addition to elevated suicide rate—when compared to their metropolitan counterparts. Rural Australians are more likely to suffer from a range of chronic health conditions and are at greater risk of accidental death (e.g., road transport accidents) than those in metropolitan settings [5]. However, rural Australia is not homogeneous, and variously comprises coastal areas with high numbers of tourists and retirees, mining communities with the accompanying transient populations, remote indigenous communities and large areas of farming and agriculturally based communities. The factors contributing to health inequities—including suicide risk and accidental death—are likely to reflect this heterogeneity and require detailed exploration to support any specific prevention and intervention strategies.

Farmers are vital for maintaining the production of food and fibre in an environment of ever-increasing populations and market demand. In Australia—as in many other Western nations—agricultural production is increasingly recognised as a vulnerable industry. Farmers are ageing, and facing increasing technological and mechanical demands, in a fluctuating global marketplace with mounting climate uncertainty [6,7,8]. Given that the majority of Australian farms remain family owned and operated [9], this occupational vulnerability is likely to extend beyond those defined as farmers to include family members living and helping out on farms.

Australia’s farmers have been identified as at risk of psychological distress [10,11] and heightened rates of suicide [12,13,14,15], in the absence of any clear evidence of higher rates of diagnosed mental illness [16]. Poor mental health and suicide risk in farming have been attributed to a complex range of interconnected cultural, environmental, geographical, social and psychological risk factors [11,17,18,19,20]. These factors include poor access to support services [21,22], an unsustainable work ethic [23], uncertainty and lack of control in farming [23], social disconnection [17,19], poor business profitability [24], acclimatisation to risk taking [25] and access to means [14]. Qualitative research suggests there may be two distinct pathways to suicide for Australia’s farmers—an acute situational pathway (reflecting risk factors associated with interpersonal relationships, financial stressors and retirement) and a protracted pathway linked with mental illness [17].

Much of the recent Australian evidence on farming-related suicide is reported from small-scale qualitative studies of farming populations in the states of Queensland and New South Wales (NSW) [17,18,19,23,26]. Interviews with community members and families bereaved by farmer suicide provide in-depth insights. However, given the diverse nature of rural communities, the generalisability of findings to farmers in other parts of Australia is limited. A larger study of farmer suicide by Arnautovska and colleagues [20,27] incorporating Queensland and NSW suggested that there is not only state differences but also significant geographic variability of suicide rates across regions within states. Further to this point, the extant literature also offers little guidance as to how farming-related suicide might differ from suicide in rural communities more broadly. While studies have identified farming-specific risk factors in suicide, Australians living outside of major metropolitan cities generally face a range of health inequalities when compared to their metropolitan counterparts (Australian Institute of Health and Welfare, 2017), which may contribute to suicide risk regardless of whether a person is engaged in farming.

Clear gender differences have also been noted in the Australian literature to date, with farming males dying at a rate 3.8 times that of farming females [27]. This reflected similar gender differences in the non-farming population, where males died by suicide at a rate 3.6 times that of females [27]. 

Victoria is Australia’s most densely populated state, covering approximately the same area as the United Kingdom with a population of 5.93 million people [4]. The state of Victoria (located in the Southeast of the country) is Australia’s largest agricultural producer ($14.9 billion) and largest food and fibre exporter, with the majority of farm businesses comprising beef and sheep, grains, dairy, and grapes [28]. To date there have not been any detailed population-level analyses of farming-related suicides in Victoria.

In this study, we compared the demographic characteristics and suicide death circumstances of farming- and non-farming-related suicides in rural Victoria. Our purpose was twofold: to improve understanding of farming-related suicide in Australia’s largest agricultural producing state; and to explore how farming-related suicides might differ—or not—from suicide in rural communities more broadly. Achieving this purpose would contribute to the development of relevant and targeted suicide prevention responses for these vulnerable populations.

## 2. Materials and Methods 

### 2.1. Study Design 

This retrospective cohort study compared farming- and non-farming-related suicide deaths of residents living in rural (defined for the purposes of this study as all locations outside of metropolitan Melbourne) areas of the state of Victoria between 1 January 2009 and 31 December 2015. 

### 2.2. Data Source

This study utilised data from the Victorian Suicide Register (VSR) at the Coroners Court of Victoria (CCOV). All deaths in Victoria from suspected non-natural causes must be reported to the CCOV for Coronial investigation. If a death occurs in circumstances consistent with suicide, it is entered into the VSR. The VSR includes a core dataset of basic information for every death (including age, sex, location of death, location of residence, and suicide method). Additionally, an enhanced dataset is progressively being coded, which encompasses detailed information about stressors, health service contacts, mental ill health, and several other domains. The enhanced dataset is coded with reference to all evidence gathered in the course of the Coroner’s investigation, such as statements of family and friends, emergency services, health professionals, witnesses and employers. At the time this study was approved, enhanced coding was completed for all suicides between 2009 and 2015.

### 2.3. Case Selection

A search of the VSR identified 1323 rural suicide deaths between 1 January 2009 and 31 December 2015 (the period for which the enhanced VSR dataset was available). Location was determined by the deceased’s usual place of residence. Twenty deaths were excluded from the study because of incomplete VSR dataset coding. Two deaths were excluded because the deceased were being held in custody in a rural prison but did not usually reside in rural Victoria. During project coding, three additional deaths were excluded due to insufficient information for locating and/or classifying the deceased’s usual place of residence. This resulted in a final study cohort of 1298 rural deaths. 

Farming-related suicides were determined by the deceased’s usual residential location (through analysis of latitude and longitude) or non-residential connection to farming (via employment). This enabled the inclusion of data for suicide deaths of people who were likely exposed to the stressors of farming work and life, even when they did not identify as farmers by occupation (i.e., suicide deaths of people who were identified by their off-farm occupation but lived on a farm and were involved in the farming business; farming family members who participated in farming life and work but did not identify as farmers by occupation [e.g., identified according to their additional off-farm employment, identified as farmer’s wives or homemakers, identified as retired]). Coding of these variables was informed by geo-codes and detailed text-based data contained in the VSR.

### 2.4. Data Compilation

Relevant data on each case were extracted from the VSR into a new, de-identified database from which further analysis could be conducted. This included demographic variables of age, sex, sexuality, ethnicity, employment, relationship status, residential Primary Health Network, residential type and whether the deceased resided in a town with a population over 1000 people. Details of the suicide location and method were recorded. The database identified contextual factors experienced by the decedents including experience of abuse, presence of stressors (work, financial, legal, sexuality, isolation, abuse, education, bullying and substance use), family and friend-related stressors (death, separation, conflict, health, violence), and previous exposure to suicide. Evidence was recorded on the decedents’ physical and mental health, specifically presence of diagnosed or suspected mental illness, and utilisation of mental health treatment services. 

### 2.5. Data Analyses 

Data were analysed using IBM Statistical Package for the Social Sciences (SPSS) version 25 (IBM, New York). Data were compared across farming-related and non-farming-related suicide cases. Farming-related suicides were identified if a case was coded as living on a commercial farm or had a non-residential connection to farming. Descriptive statistics were used to describe the variables. Frequencies and percentages were calculated to compare suicide method, presence and diagnosis of mental illness, substance use, mental health treatment, presence of life events and stressors, and experience of abuse. Inferential statistics were used to compare the farming-related suicide and non-farming populations. Chi-squared tests were used to compare farming- and non-farming-related cases to identify risk factors for suicidal deaths. Statistical significance was determined by *p* < 0.05. Strength of association between suicidal deaths and risk factors was determined by odds ratio (ORs) through logistic regression analyses; 95% confidence intervals (CIs) were also calculated. Multivariable analyses were conducted to control potential confounding factors; adjusted ORs with 95% CIs were calculated. 

### 2.6. Ethics

Ethical approval was obtained from Deakin University HREC (No. 2018-224). Victoria’s State Coroner endorsed access to VSR data for the purposes of this study. 

## 3. Results

### 3.1. Demographic Characteristics

This study examined the 1298 fully-coded non-metropolitan suicide deaths identified in the VSR. This sample was divided into farming-related suicides (N = 133) and non-farming-related (N = 1165) suicides. The mean age of the sample was 47 years (±17.937), with 81% (N = 1049) male. Of the total sample, 73% (N = 943) resided in a town with a population over 1000. Specifically, 81% (N = 1057) lived in an environment with neighbours in close proximity (determined by geocode), 9% (N = 116) a rural lifestyle property, 9% (N = 115) on a commercial farm and 1% (N = 10) were classified as other (insufficient information available to determine residential status). 

There was no significant difference (*p* > 0.05) in the mean age between farming-related (48 (±18.5) years) and non-farming-related (47 (±17.9) years) suicide deaths. Table 1 shows the demographic characteristics of farming-related and non-farming-related suicide deaths. Suicide was more common amongst males within both cohorts (farming-related: 82% male, non-farming-related: 81% male). There was no significant difference in the proportion of male suicides (*p* > 0.05) between farming- and non-farming-related deaths.

People who reported farming-related suicides were more likely to be employed (n = 70, 53%) (*p* < 0.001) compared to their non-farming-related counterparts (n = 439, 38%) (OR 1.84, 95% CIs 1.28–2.64). There was no statistically significant difference (*p* > 0.05) between the two groups on sexuality or relationship status. 

### 3.2. Circumstances of Suicide

Table 2 describes the mechanisms and circumstances of death by suicides in rural areas. Hanging and asphyxia was the most common method for both farming-related (n = 54, 41%) and non-farming-related (n = 625, 54%) suicides. However, firearms were used significantly more often (*p* < 0.001) as a means of farming-related suicide (n = 40, 30%) than non-farming-related deaths (n = 101, 9%). For deaths by firearm, shotguns (long-arm rifles) were most commonly used (n = 102, 72%). Poisonings (drug and non-drug related poisoning, engine exhaust gassing and irrespirable atmospheres) were significantly (*p* < 0.05) higher for non-farming-related (n = 306, 26%) than farming-related suicide deaths (n = 24, 18%). 

Forty-five per cent of all suicide decedents had a previously diagnosed mental health condition, with non-farming-related deaths (n = 537, 46%) more likely (*p* < 0.05) to be diagnosed than farming-related suicide deaths (n = 48, 36%). Non-farming-related deaths were more likely (*p* < 0.05) to have used substances. Whilst not statistically significant (*p* > 0.05), farming-related suicides (n = 95, 71%) were more likely suspected of having a mental illness when compared to non-farming-related deaths (n = 733, 63%). 

Fewer than 40% of all cases had received mental health treatment within the six weeks before death. Non-farming-related deaths (n = 583, 50%) were more likely (*p* < 0.05) to have received mental health treatment more than six weeks before death compared to farming-related suicide deaths (n = 53, 40%). 

There were no significant differences between farming-related and non-farming-related suicide deaths in relation to the presence of stressors or interpersonal problems. In both groups the presence of a physical health problem (n = 575, 44%) was the most commonly identified stressor. In both groups, abuse (as either victim or perpetrator) was identified in approximately one out of every five suicide deaths.

### 3.3. Diagnosed and Suspected Mental Health Illnesses 

Table 3 shows the presence of diagnosed and suspected mental illness within all suicide deaths. Mood disorders were the most commonly diagnosed disorder for both the farming-related (n = 33, 25%) and non-farming-related deaths (n = 416, 36%). However, non-farming-related deaths were more likely to be diagnosed with a mood disorder than farming-related deaths (*p* < 0.05). 

Multivariate analyses showed that farming-related suicides were four times more likely to involve the use of firearms as a mechanism of suicide when compared to non-farming-related suicides (AORs 3.48, 95% CIs 1.73–7.01, *p* = 0.000). Additionally, farming-related suicides were twice as likely to demonstrate evidence of stress directly related to their legal situation (AORs 1.75, 95% CIs1.13*–*2.72, *p* = 0.01). 

### 3.4. Access to Mental Health Treatment

Table 4 shows the mental health treatment services accessed by all suicide deaths. General Practitioners (GPs) were the most utilised mental health treatment service, both within six weeks and greater than six weeks prior to death. This was identified for both farming-related and non-farming-related suicide deaths. However, farming-related deaths (n=31, 23%) were less likely to have received mental health treatment from a GP more than six weeks prior to death compared to non-farming-related deaths (n=390, 34%) (p<0.05). Drug and alcohol services and the Crisis Assessment and Treatment Team were the least used mental health services, by both cohorts, over any period prior to death. 

For suicide deaths with a previous diagnosis of mental illness, 71% (n = 34) of farming-related and 63% (n = 337) (*p* > 0.05) of non-farming-related deaths had received mental health treatment within six weeks of death. Of those suspected of having a mental health condition, 37% (n = 14) of farming-related and 41% (n = 175) (*p* > 0.05) of non-farming-related deaths had received treatment within the six weeks. 

## 4. Discussion

To address the gap in knowledge about farming-related suicide in Victoria, exploring and comparing the demographic characteristics and suicide death circumstances of farming- and non-farming-related suicides in non-metropolitan Victoria, this study draws on evidence from the Coroners Court of Victoria’s Victorian Suicide Register.

Farming-related and non-farming-related suicide deaths varied on a number of factors including sex, employment status; mechanism used to die by suicide; diagnosis, suspicion and nature of a mental health condition; evidence of substance use; evidence of legal stressors; history of mental health treatment; and, mental health treatment by a GP. Each of these will be addressed in turn.

Farming-related suicide deaths were significantly more likely to be males than females, a trend similarly reflected in the general Australian population [2]. While possible explanations have been made for this gender paradox in the general population—including male’s choice of more lethal suicide method when compared to females [29]—there is limited understanding of how the farming context may influence this gender paradox. Previous Australian research has demonstrated the exhibition of traditional masculinist behaviours (i.e., goal-directed behaviours as opposed to ‘seeking help’) in both male and female farming family members. How this directly translates into risk factors and/or circumstances of female farming-related suicide remains unknown. While the small sample size of females in this study limits meaningful quantitative comparisons, further exploration of the qualitative data contained within the Victorian Suicide Register is warranted to improve understanding of female farming-related suicide and how this may vary (or not) from male farming-related suicide.

Farming-related suicide deaths were more likely to be employed than non-farming-related suicides. This may be due to the fact that the majority of Australian farms continue to be family owned. Job opportunity and security may be greater for those involved in a family farm, even in the face of crisis or significantly reduced income. Family members may also receive support to stay within a family business, even if situational factors limit their capacity to perform their role. In contrast, non-farming-related suicide deaths may have greater exposure to traditional employment models (e.g., hired as an employee for wages/salary) and so must seek employment in a competitive market, with potentially reduced job security should they be unable to meet the requirements of their job description due to situational or mental health challenges. This is consistent with previous research identifying family businesses as more likely to maintain jobs during crisis periods than non-family businesses, even with reduced turnover—an outcome attributed to their ownership and management characteristics which prioritise affective needs such as identity, ability to examine influence, status and continuation of the family dynasty [30,31]. Higher identified levels of employment risk obscuring the employment-related stressors that farming-related suicides may have been exposed to, such as reduced income. This is supported by the data identifying farming-related suicides as experiencing financial stress at the same rate as non-farming-related suicides, despite the differences in job-related stress. A more detailed understanding of the complexities of financial and job-related stress are required to understand suicide risk in farming populations.

While firearms were significantly more likely to be used as a means of suicide for farming- than non-farming-related deaths, hanging remains the most likely means of suicide. Higher rates of firearm suicide in the farming population are consistent with the fact that firearms (particularly long arms) are commonly used as a farming tool [17]. While restriction of means is frequently discussed as a way to reduce suicide rates—particularly in relation to firearms [32,33]—this approach has its limitations. Firstly, restricting access to firearms for farmers can be problematic over a longer term by reducing the capacity to operate the farming business. Secondly, restricting access to means for hanging—the most common means of suicide in farming- and non-farming-related suicides—is particularly difficult, given the ready access to everyday items such as ropes, cords, chain, etc., required on a rural property and easily accessible in town centres. Thirdly, focusing on means—a factor that is likely to have a moderating, rather than a causal effect on suicide—should not discount the complex range of factors contributing to risk, preceding any consideration of means. Instead, prevention efforts should concentrate on ensuring appropriate ownership, licensing and safe storage of firearms whilst also reducing exposure to farming-related stress factors, building personal capacity to withstand stressors and providing access to appropriate support and mental health treatment services.

Farming-related deaths were more likely to be associated with an identified legal stressor. This confirms the presence of legal factors as a factor in farmer suicide identified (but not detailed) in previous qualitative research [17]. While the quantitative data of the Victorian Suicide Register does not identify the nature of these legal stressors, further research should conduct a deeper exploration of the qualitative data contained within the Register to develop a clearer understanding of what these legal stressors comprise (e.g., personal legal issues or issues associated with the farming business). Detailed knowledge about these legal stressors would help inform targeted prevention efforts.

Farming-related deaths were both less likely to have received a mental health diagnosis and (therefore, understandably) less likely to have received mental health treatment in the six weeks prior to their death. Over the longer term, farming-related suicide deaths were also less likely to seek treatment from their General Practitioner (possibly the only provider of health services in a rural area) than non-farming-related deaths. While people in rural areas generally have less access to mental health services [22], the increasing size and reducing numbers of farms may further contribute to geographic isolation and limited service access [34]. In addition to reduced access to services is the limitation and possible effect posed by inappropriate service provision. For support services to be considered appropriate by farmers, they must demonstrate an understanding of farming life and work [35]. Lack of cultural competency by health professionals can contribute to a reluctance to access services [36]. Compounding this, farmers demonstrate a pragmatic, goal-focused attitude to facing challenges, and a tendency to volunteer help to others while avoiding seeking help themselves [25]. Further work is required to build cultural competence in our rural health workforce and develop pathways to treatment and support which are both accessible and appropriate for our farming populations.

There was a concerning proportion of rural suicide deaths (farming- and non-farming-related) where abuse (as perpetrator or victim) was identified as a contributing factor. Evidence has consistently identified domestic and family violence as occurring at higher rates in Australia’s rural areas [37], with some of the highest rates identified in agricultural communities [38]. Stigma, lack of anonymity, geographic isolation, complex financial arrangements, poor access to services and exposure to natural disaster all contribute to the occurrence and outcomes related to domestic and family violence in rural Australia [37]. Violence has been clearly linked to suicide, with a recent international meta-analysis highlighting links between the experience of child abuse and suicide attempt in later life [39]. Further investigation should endeavour to better understand the nature and direction of the abuse noted in the Victorian Suicide Register, with a view to designing appropriate tailored prevention and support strategies in rural areas.

### 4.1. Strengths and Limitations

This study has significant strengths, including addressing challenges identified in previous farmer suicide research [40]. Firstly, this is the first study in Australia to actually compare farming-related suicides with other rural suicides (as opposed to modelling), and draws on a large, state-wide sample from across a 7 year period. Secondly, this study looked beyond those defined by occupation to also include suicide deaths of people residing on farms (including farming family members)—a population who are likely to be exposed to many of the stressors of farming life and work. There are also several limitations to this study. Firstly, coronial data are not gathered for the purpose of research and are, therefore, varied in detail and consistency. Secondly, non-metropolitan Victorian suicides include all deaths outside of metropolitan Melbourne and, therefore, include deaths from varied locations ranging from large regional towns to remote areas which may have varied access to support services not captured here. Thirdly, the time period which this data spans is unlikely to represent the full flow-on effect of farming-related stressors experienced from 2009 to 2015. Fourthly, given the focus of those who died by suicide, this study is unable to shed light on the protective factors available to people similarly exposed to farming-related stressors who successfully navigated these challenges. Finally, the capacity to extrapolate the findings from this study beyond the state of Victoria is limited, given the previously identified regional variability of farming-related suicide [20,27]. 

### 4.2. Implications for Prevention

Prevention efforts should focus on influencing a wide range of factors to reflect the complexity of suicide risk identified in this research. This supports the current implementation across Australia trialing an integrated systems approach to suicide prevention. For farming communities, this may include:
Adopting a broader approach to fostering working conditions supportive of the health, well-being and safety of farm owners, managers, workers and farming families. Such an approach would need to encourage positive conditions in which employment occurs (including access to training, adequate remuneration, effective succession planning and safe working environment).Developing policy, legislation and training to ensure firearms owned are appropriate for the required task and safely stored, and provide opportunity for at-risk community members (or their family members) to proactively transfer the possession of firearms from those at risk during periods of situational risk, without long-term punitive consequences which may hinder the capacity to effectively farm.Developing a range of support services that are culturally appropriate and accessible to farming community members. This includes halting the ongoing diminishment of locally-available services in rural communities, and develop new complementary service models including outreach, online and phone services and developing culturally competent, skilled health professionals and peer networks for the delivery of health, well-being and safety support.

Many factors considered in this study were identified at similar rates for farming- and non-farming suicide deaths. This suggests prevention efforts must also consider the flow-on effect of farming-related stress on to the broader rural community, including rural service providers and small businesses.

## 5. Conclusions

This study compared demographic characteristics and suicide death circumstances of farming- and non-farming-related suicides in rural Victoria with the aim of: a) exploring the contributing factors to farming-related suicide in Australia’s largest agricultural producing state; and b) examining whether farming-related suicides differ from suicide in rural communities. There were many similarities in the contributing factors and characteristics of rural suicide deaths irrespective of farming involvement. However, farming-related suicide deaths were more likely to be employed at the time of death and have died through use of a firearm. In addition, farming-related suicides were less likely to have a diagnosed mental illness or have received mental health support more than six weeks prior to death. A range of suicide prevention strategies need adopting across all segments of the rural population irrespective of farming status. However, data from farming-related suicides highlight the need for targeted firearm-related suicide prevention measures and appropriate, tailored and accessible support services to support health, well-being and safety for members of farming communities.

## Figures and Tables

**Table 1 ijerph-17-02010-t001:** Demographics of Suicide Deaths, Farming- and Non-Farming.

	Total Sample (N = 1298)	Farming (n = 133)	Non-Farming (n = 1165)	
	n (%)	n (%)	n (%)	OR (95% CI)
**Sex**				
Male	1049 (80.8%)	109 (82.0%)	940 (80.7%)	0.92 (0.58–1.47)
Female	249 (19.2%)	24 (18.0%)	225 (19.3%)	
**Sexuality/Gender**				
Heterosexual	1262 (97.2%)	129 (97.0%)	1133 (97.3%)	1.10 (0.38–3.15)
LGBTI	36 (2.8%)	4 (3.0%)	32 (2.7%)	
**Ethnicity**				
Non-Indigenous	1269 (97.8%)	133 (100.0%)	1136 (97.5%)	
Indigenous	29 (2.2%)	0 (0.0%)	29 (2.5%)	
**Employment**				
Employed	509 (39.2%)	70 (52.6%)	439 (37.7%)	1.84 (1.28–2.64) **
Unemployed/Unable to Work	408 (31.4%)	26 (19.5%)	382 (32.8%)	0.50 (0.32–0.78) **
Retired/Pensioner	246 (19.0%)	24 (18.0%)	222 (19.1%)	0.94 (0.59–1.49)
Other	135 (10.4%)	13 (9.8%)	122 (10.5%)	0.93 (0.51–1.69)
**Relationship Status**				
Not in a Relationship	756 (58.2%)	72 (54.1%)	684 (58.7%)	0.83 (0.58–1.19)
In a Relationship	542 (41.8%)	61 (45.9%)	481 (41.3%)	1.21 (0.84–1.73)

** *p* < 0.01.

**Table 2 ijerph-17-02010-t002:** Mechanism and Circumstances of Farming-Related and Non-Farming-Related Suicide Deaths.

	Total Sample (N = 1298)	Farming Related (n = 133)	Non-Farming Related (n = 1165)	
	n (%)	n (%)	n (%)	OR (95% CI)
**Mechanism**				
Firearm	141 (10.9%)	40 (30.1%)	101 (8.7%)	4.51 (2.97–6.92) ***
Longarm/shotgun	102 (72.3%)	32 (24.1%)	70 (6.0%)	1.77 (0.73–4.28)
Handgun	4 (2.8%)	0 (0.0%)	4 (0.3%)	
Unknown	35 (24.8%)	8 (6.0%)	27 (2.3%)	0.69 (0.28–1.67)
Hanging and asphyxia	679 (52.3%)	54 (40.6%)	625 (53.6%)	0.59 (0.41–0.85) **
Poisoning	330 (25.4%)	24 (18.0%)	306 (26.3%)	0.61 (0.39–0.98) *
Sharp object	23 (1.8%)	2 (1.5%)	21 (1.8%)	0.83 (0.19–3.59)
Jump from height/impact of vehicle	81 (6.2%)	8 (6.0%)	73 (6.3%)	0.96 (0.45–2.03)
Other	44 (3.4%)	5 (3.8%)	39 (3.3%)	1.13 (0.44–2.91)
**Evidence of Mental Health and Substance Use**				
Mental illness diagnosed	585 (45.1%)	48 (36.1%)	537 (46.1%)	0.66 (0.46–0.96) *
Mental illness suspected	828 (63.8%)	95 (71.4%)	733 (62.9%)	0.68 (0.46–1.01)
Substance use	578 (44.5%)	46 (34.6%)	532 (45.7%)	0.63 (0.43–0.92) *
Received mental health treatment within six weeks of death	500 (38.5%)	44 (33.1%)	456 (39.1%)	0.77 (0.53–1.12)
Received mental health treatment more than six weeks before death	636 (49.0%)	53 (39.8%)	583 (50.0%)	0.66 (0.46–0.95) *
**Evidence of Life Events/Stressors**				
Physical health problem	575 (44.3%)	60 (45.1%)	515 (44.2%)	1.04 (0.72–1.49)
Job problem	378 (29.1%)	29 (21.8%)	349 (30.0%)	0.65 (0.42–1.00)
Financial problem	388 (29.9%)	39 (29.3%)	349 (30.0%)	0.97 (0.65–1.44)
Legal stressor	316 (24.3%)	39 (29.3%)	277 (23.8%)	1.33 (0.89–1.98)
Sexuality identification stressor	28 (2.2%)	4 (3.0%)	24 (2.1%)	1.47 (0.50–4.32)
Isolation stressor	174 (13.4%)	11 (8.3%)	163 (14.0%)	0.55 (0.29–1.05)
Education stressor	43 (3.3%)	3 (2.3%)	40 (3.4%)	0.65 (0.20–2.13)
**Evidence of Interpersonal Problems**				
Experience of bullying	148 (11.4%)	17 (12.8%)	131 (11.2%)	1.16 (0.67–1.99)
Witness of abuse	15 (1.2%)	2 (1.5%)	13 (1.1%)	1.35 (0.30–6.06)
Perpetrator of abuse	255 (19.6%)	26 (19.5%)	229 (19.7%)	0.99 (0.63–1.56)
Victim of abuse	219 (16.9%)	24 (18.0%)	195 (16.7%)	1.10 (0.69–1.75)
Suicide of family member/partner	126 (9.7%)	9 (6.8%)	117 (10.0%)	0.65 (0.32–1.31)
Suicide of friend	29 (2.2%)	2 (1.5%)	27 (2.3%)	0.64 (0.15–2.73)

* *p* < 0.05; ** *p* < 0.01; *** *p* < 0.001.

**Table 3 ijerph-17-02010-t003:** Diagnosed and Suspected Mental Health Disorders.

	Total Sample (N = 1298)	Farming Related (n = 133)	Non-Farming Related (n = 1165)	
	n (%)	n (%)	n (%)	OR (CI 95%)
**Diagnosed**				
Organic mental disorders	21 (1.6%)	0 (0.0%)	21 (1.8%)	
Mental and behavioural disorders due to psychoactive substance abuse	104 (8.0%)	10 (7.5%)	94 (8.1%)	0.93 (0.47–1.83)
Schizophrenia, schizotypal and delusional disorders	74 (5.7%)	6 (4.5%)	68 (5.8%)	0.76 (0.32–1.79)
Mood disorder	449 (34.6%)	33 (24.8%)	416 (35.7%)	0.59 (0.39–0.90) *
Neurotic, stress-related and somatoform disorders	169 (13.0%)	13 (9.8%)	156 (13.4%)	0.70 (0.39–1.27)
Behavioural symptoms associated with physiological disturbances and physical factors	21 (1.6%)	3 (2.3%)	18 (1.5%)	1.47 (0.43–5.06)
Disorders of adult personality and behaviour	56 (4.3%)	9 (6.8%)	47 (4.0%)	1.73 (0.83–3.61)
Mental retardation	4 (0.3%)	0 (0.0%)	4 (0.3%)	
Psychological development disorders	9 (0.7%)	0 (0.0%)	9 (0.8%)	
Behavioural and emotional disorders with onset usually occurring in childhood	19 (1.5%)	2 (1.5%)	17 (1.5%)	1.03 (0.23–4.51)
Unspecified mental disorders	2 (0.2%)	0 (0.0%)	2 (0.2%)	
**Suspected**				
Organic mental disorders	15 (1.2%)	1 (0.8%)	14 (1.2%)	0.62 (0.8–4.77)
Mental and behavioural disorders due to psychoactive substance abuse	210 (16.2%)	12 (9.0%)	198 (17.0%)	0.48 (0.26–0.89) *
Schizophrenia, schizotypal and delusional disorders	19 (1.5%)	0 (0.0%)	19 (1.6%)	
Mood disorder	231 (17.8%)	27 (20.3%)	204 (17.5%)	1.20 (0.77–1.88)
Neurotic, stress-related and somatoform disorders	48 (3.7%)	4 (3.0%)	44 (3.8%)	0.79 (0.28–2.23)
Behavioural symptoms associated with physiological disturbances and physical factors	9 (0.7%)	1 (0.8%)	8 (0.7%)	1.10 (0.14–8.83)
Disorders of adult personality and behaviour	48 (3.7%)	2 (1.5%)	46 (3.9%)	0.37 (0.09–1.55)
Mental retardation	0 (0.0%)	0 (0.0%)	0 (0.0%)	
Psychological development disorders	6 (0.5%)	1 (0.8%)	5 (0.4%)	1.76 (0.20–15.16)
Behavioural and emotional disorders with onset usually occurring in childhood	5 (0.4%)	0 (0.0%)	5 (0.4%)	
Unspecified mental disorders	5 (0.4%)	1 (0.8%)	4 (0.3%)	2.20 (0.24–19.82)

* *p* < 0.05.

**Table 4 ijerph-17-02010-t004:** Mental Health Treatment Services Accessed Prior to Death.

	Total Sample (N = 1298)	Farming Related (n = 133)	Non-Farming Related (n = 1165)	
	n (%)	n (%)	n (%)	OR (CI 95%)
**Proximal treatment (within six weeks)**				
Psychiatrist	183 (14.1%)	19 (14.3%)	164 (14.1%)	1.02 (0.61–1.70)
Psychologist	78 (6.0%)	5 (3.8%)	73 (6.3%)	0.58 (0.23–1.47)
Mental health professional	188 (14.5%)	18 (13.5%)	170 (14.6%)	0.92 (0.54–1.55)
General practitioner	310 (23.9%)	25 (18.8%)	285 (24.5%)	0.72 (0.45–1.13)
Emergency department	96 (7.4%)	6 (4.5%)	90 (7.7%)	0.56 (0.24–1.32)
Crisis Assessment and Treatment Team	29 (2.2%)	2 (1.5%)	27 (2.3%)	0.64 (0.15–2.74)
Drug and alcohol service	30 (2.3%)	1 (0.8%)	29 (2.5%)	0.30 (0.04–2.20)
**Treatment at other time**				
Psychiatrist	225 (17.3%)	18 (13.5%)	207 (17.8%)	0.72 (0.43–1.22)
Psychologist	112 (8.6%)	9 (6.8%)	103 (8.8%)	0.75 (0.37–1.52)
Mental health professional	186 (14.3%)	15 (11.3%)	171 (14.7%)	0.74 (0.42–1.30)
General practitioner	421 (32.4%)	31 (23.3%)	390 (33.5%)	0.60 (0.40–0.92) *
Emergency department	85 (6.5%)	4 (3.0%)	81 (7.0%)	0.42 (0.15–1.51)
Crisis Assessment and Treatment Team	33 (2.5%)	2 (1.5%)	31 (2.7%)	0.56 (0.13–2.36)
Drug and alcohol service	35 (2.7%)	1 (0.8%)	34 (2.9%)	0.25 (0.034–1.86)

* *p* < 0.05.

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
