# Peer review of "Suicide in Rural Australia: Are Farming-Related Suicides Different?"

_ijerph, 2020, doi:10.3390/ijerph17062010_

Round 1

Reviewer 1 Report

Aim: The purpose of this paper is to describe the contributing factors to farming-related suicide in Victoria Australia compared to rural non-farming related suicides. The overall purpose is to assess whether farm related suicide and non-farm rural suicide differ in circumstances or means.

Main contributions: The primary contribution of this paper to the literature on suicide is the ability they authors had to assess detailed circumstances related to the suicide. Specifically, the database they used provided information related to evidence of mental health and substance use, evidence of life events/stressors, and evidence of interpersonal problems. This information is a strength of the overall paper and provides an in-depth comparison between farm related and nonfarm related rural suicide.

Analysis: The approach to analysis was appropriate and provided information regarding the circumstances and contributors to suicide in the rural area with a robust ability to compare farm and nonfarm circumstances. Results: The results of this paper provide information that can be used to focus prevention and intervention programs specific for rural farm and rural nonfarm communities. The authors properly identify those issues. Firearm safety may play a significant factor in the completion of suicide among farm residents. Additionally, farm residents were less likely to b diagnosed with a mood disorder compared to other rural residents. Further, farm residents were more likely to have evidence of stress related to their legal situation compared to other rural residents. In contrast, rural residents were more likely to be unemployed compared to farm residents. Clearly targeting unemployed workers in the rural community and firearm safety in the farming population may be important prevention/interventions strategies.

Strengths of the study: The study included a large sample with detailed information regarding circumstances and methods of suicide in a rural population that allowed a comparison of farm and non-farm residents. The authors were able to include farm residents as well as those employed in farming which allowed for the inclusion of a broader sample of farm related suicides than previous studies relying solely on occupational codes.

Limitations of the study: The limitations are related to the use of death certificates, meaning that there is no ability to assess protective factors and to assess the relative importance of those in relation to factors that are associated with completed suicide. Nonetheless, the paper contributes to the literature on farm suicide in important ways as noted above. Minor edit: Although the authors note that legal situations differed in the association between suicide in farm and non-farm residents, they do not provide an explanation/definition of what types of legal situations are included in this assessment. It would be helpful to clarify what might be included as this may provide another venue for developing prevention/intervention programs targeted at the farming populations at risk for suicide.

Conclusion: The paper makes an important contribution to the literature on suicide among farm populations but also provides additional information on rural suicide that may be useful in developing targeted interventions to reduce suicide risk in both groups while addressing specific characteristics that lead to high risk of suicide.

Author Response

Reviewers comment:

Minor edit: Although the authors note that legal situations differed in the association between suicide in farm and non-farm residents, they do not provide an explanation/definition of what types of legal situations are included in this assessment.

Author response:

The authors agree that further investigation as to the detailed nature of legal stressors would be valuable for prevention activities. Although beyond the scope of this paper, a deeper investigation of the available qualitative data in the 'notes' section of the Victorian Suicide Register on stressors of interest is planned by the authors. This has been acknowledged and added to the  manuscript discussion (lines 268 - 274).

Reviewer 2 Report

Internationally, Australia contributes a large proportion of the literature on farmer distress and suicide and is continually leading new advances in knowledge in this field. Within this field, studies drawing on data from suicide fatalities are fairly infrequent but much needed to inform suicide prevention policy, programs and initiatives. The present manuscript offers original and significant analyses of coronial data from the Victorian Suicide Register that will be welcomed by researchers and stakeholders in farmer suicide prevention in Australia and throughout the western nations that share disproportionately high rates of suicide for farmers and similar risk profiles. 

The present manuscript utilizes the suicide fatality data for rural Victoria to take up the question of whether and how farming-related suicides differ from non-farming suicides to identify factors that particularly characterize farming-related suicides in order to inform targeted and tailored intervention and prevention efforts.

The literature review succinctly canvases the factors for farmers that contribute to risk of suicide (pg 2) and reinstates the significant point that for farmers, heightened rates of suicide occur in the absence of any clear evidence of higher rates of diagnosed mental illness. Here, it would be worth briefly mentioning the recent study by Kunde et al (2017, this journal) that points to two distinct pathways to suicide for Australian farmers - an acute situational pathway and a protracted pathway characterized by long term mental illness. In addition, the most significant risk factor for farmers is not identified or discussed - and that is gender. Of course, it is ubiquitously understood that suicide is gendered and that men and male farmers comprise the majority of their respective suicide fatality statistics. The findings of the current study once again demonstrate that point - but to overlook gender in the introduction and analysis risks normalising its contribution and erasing its significance for understanding suicide. In addition, very little is known about the suicide deaths of women in farming, or rural women, for that matter. The study reports 24 suicide fatalities for women in the data set but no further analysis of risk factors by sex. This may be a missed opportunity to add further novel insights to the paper - unless inferences are constrained by the small sample size in which case perhaps separate analysis of the data on rural female deaths may be possible in another paper.  

Lines 114-118 were confusing as they seemed repetitive.

In the findings (pg 6) the high percentages pertaining to abuse caught my eye and may warrant discussion. They may not be statistically different between groups but represent an important line for further inquiry and intervention / support.

Given that alcohol consumption is often identified as playing a mediating role in suicide deaths I wondered whether BAC is recorded in the data set and could be included in the analysis?

Line 248 typo ‘faming’

Line 337-338 Looks like the instructions have been included in the manuscript – should this line be deleted?

Author Response

1. Reviewer comment:

it would be worth briefly mentioning the recent study by Kunde et al (2017, this journal) that points to two distinct pathways to suicide for Australian farmers - an acute situational pathway and a protracted pathway characterized by long term mental illness.

1. Author response:

While the Kunde study was referenced in the introduction, it has now been further detailed to include the two identified pathways to suicide (line 63-66).

2. Reviewer comment:

the most significant risk factor for farmers is not identified or discussed - and that is gender.

2. Author response:

The author's agree that gender is a very important point to note--in the introduction, analysis and discussion. Additions have been made to reflect this:

Introduction: lines 80-82

Discussion: lines 240-251

3. Reviewer comment:

Lines 114-118 were confusing as they seemed repetitive.

3. Author response:

The author's agree that this sounds repetitive and we have removed some of the text.

4. Reviewer comment:

In the findings (pg 6) the high percentages pertaining to abuse caught my eye and may warrant discussion. They may not be statistically different between groups but represent an important line for further inquiry and intervention / support

4. Author response:

The authors agree that this is an area that requires further understanding and prevention efforts for all rural populations (farming and non-farming). An additional paragraph has been added to the discussion to express this.

5. Reviewer comment:

Given that alcohol consumption is often identified as playing a mediating role in suicide deaths I wondered whether BAC is recorded in the data set and could be included in the analysis?

5. Author response:

Unfortunately, our data set only identified whether toxicology tests for alcohol had been conducted (bivariable yes/no) and did not provide specific information on BAC.

6. Reviewer comment:

Line 248 typo ‘faming’

6. Author response:

Typo has been adjusted to 'farming'

7. Reviewer comment: Line 337-338 Looks like the instructions have been included in the manuscript – should this line be deleted?

7. Author response: Line 337-338 has been deleted as per response to comment 3 above